# The impact of biological sex in peripheral nerve blockade: A prospective pharmacodynamic, pharmacokinetic and morphometric study in volunteers

**Markus Zadrazil**[1]*, **Peter Marhofer**[1], **Malachy Columb**[2], **Philipp Opfermann**[1], **Werner Schmid**[1], **Daniela Marhofer**[1], **Thomas Stimpfl**[3], **Sabine Reichel**[3], **Valentin Al Jalali**[4], **Markus Zeitlinger**[4]

1 Department of Anesthesia and Intensive Care Medicine, Medical University of Vienna, Vienna, Austria, 2 Manchester University Hospitals NHS Foundation Trust, Wythenshawe Hospital, Manchester, United Kingdom, 3 Clinical Department of Laboratory Medicine, Medical University of Vienna, Vienna, Austria, 4 Department of Clinical Pharmacology, Medical University Vienna, Vienna, Austria

* markus.zadrazil@meduniwien.ac.at

**Data Availability Statement:** All relevant data are within the manuscript and its Supporting Information files.

## Abstract

### Study objective

The impact of biological sex in peripheral regional anaesthesia is largely unknown. We therefore designed a prospective study in volunteers to investigate the impact of biological sex on pharmacodynamic, pharmacokinetic and morphometric characteristics for peripheral nerve blockade.

### Methods

The initial study plan was powered to include 90 volunteers to find a difference of 35 min in duration of sensory block (primary outcome variable) with 80% power and alpha error at 5%. After discussions in ethical review, a pilot study of 2 x 12 volunteers from each sex were studied. Female and male volunteers received ultrasound guided nerve blockade with 3.0 mL ropivacaine 7.5 mg mL$^{-1}$. Sensory duration of blockade, as the primary outcome, was evaluated by pinprick testing. Secondary outcomes were sensory onset time of blockade, pharmacokinetic characteristics and the visibility of ulnar nerves using ultrasound. Analyses included Mann-Whitney $U$-statistics with $P<0.05$ (two-sided) as significant.

### Results

After 24 participants, the median (IQR) duration of sensory blockade was 450 (420; 503) min in women and 480 (450; 510) min in men ($P = 0.49$). Sensory onset time of blockade, and ultrasound visibility of nerves were also similar between the study groups. The total drug exposure across time (AUC$_{0-infinity}$) was significantly higher in women ($P = 0.017$). After a the planned power re-analysis after these 24 study paticipants, which suggested that > 400 subjects would be required with 80% power and alpha error of 5% to find significance

**Funding:** The author(s) received no specific funding for this work.

**Competing interests:** The authors have declared that no competing interests exist.

for the primary outcome parameter for marginal differences, we terminated the study at this point.

## Conclusions

We did not detect significant differences between female and male study participants in terms of pharmacodynamic and morphometric characteristics after ultrasound guided ulnar nerve blocks. Women did show significantly greater pharmacokinetic ropivacaine exposures. The results of this study indicate that peripheral regional block pharmacodynamic characteristics are independent of the biological sex, whereas pharmacokinetic parameters are sex-dependent.

## Introduction

Personalized medicine is an essential development in the individual treatment of patients and is based on tailored medical decisions, practices and interventions. Biological sex is one important element of personalized medicine.

Regional anesthesia plays a central role in perioperative medicine, thus providing the highest level of pain therapy during and after surgery. The use of ultrasound for the detection of peripheral nerves, the adjacent anatomy, the cannula and the spread of local anesthetic solutions increased the efficacy of regional anesthesia significantly and contributed to the reliability of regional anesthesia [1, 2].

Until today, the focus of scientific efforts in regional anesthesia was mainly based on the optimization of blockade techniques, the evaluation of pharmacodynamic characteristics of regional anesthetic blocks and the investigation of pharmacokinetic data of local anesthetics and additive pharmaceuticals. Knowledge of sex-differences in regional anesthesia with respect to duration or onset of regional blocks, to refer only two regional block characteristics, may extend its practical applications, but is limited with controversial results. One clinical study showed prolonged blockade of the sciatic nerve in women compared with men with pre-existing diabetes [3]. Sex-differences in anatomical characteristics of the various nerves were shown by another study [4]. And Fujumaki et al. showed differences of sensory nerve action potential of upper limb nerves between males and females [5]. Conversely, Cullion et al. and Gaudreault et al. did not find differences of pharmacodynamic characteristics of nerve blocks in rats [6, 7].

Until today, no study investigated sex-specific pharmacodynamic characteristics of peripheral nerve blocks in humans. We therefore designed this single-centre, prospective study in volunteers to investigate the influence of biological sex (men and women) on ultrasound guided peripheral nerve blockade.

## Materials and methods

### Trial authorisation

We obtained approval of the study protocol, which is available as supporting information (S1 Protocol. Study protocol), from the institutional review board (ethics commission) at the Medical University of Vienna (ref. 1787/2022, approved Dec. 22$^{nd}$ 2022) and registered the study at the European Union Drug Regulating Authorities Clinical Trials (EudraCT, ref. 2022-003201-30) and the German Clinical Trial Register (DRKS, ref. 00030404).

## Study design and participants

We recruited female and male volunteers aged 18–55 years with a body mass index of 18–35 kg m$^{-2}$ to receive one ultrasound guided ulnar nerve blockade. The definition of sex was determined by self-report. The volunteers were recruited via the Department of Clinical Pharmacology of the Medical University of Vienna and paid according to the legal standards for payment of volunteers for clinical studies. Study participants were included after written informed consent. The recruitment period lasted from January 18, 2023 to February 14, 2023. Exclusion criteria were hypersensitivity or allergy to the study drugs or poor visibility of the ulnar nerve upon ultrasound at the projected puncture site (see below). Collected data are reported according to the TREND Statement Checklist, which is attached in S1 Table.

## Ulnar nerve blockade

All volunteers received an ultrasound guided ulnar nerve block at the upper third of the non-dominant forearm. All nerve blocks were performed by the same investigator (PM) to minimize block variability. After surgical disinfection, the ulnar nerve was visualized between the flexor carpi ulnaris, humeroulnar head of the superficial flexor digitorum and profound flexor digitorum muscles with a high-resolution ultrasound system (SonoSite X-Port™, Fujufilm SonoSite Inc., Bothell, WA, USA) and a 15 MHz linear ultrasound probe. The nerve block was performed via an in-plane needle guidance technique with a 50 mm Facette tip cannula (Polymedic™, te me na, Carrières sur Seine, France). Three mL ropivacaine 0.75%wt/vol (= 22.5 mg) were administered extra-epineurally to the ulnar nerve. Male and female participants received the same dose of ropivacaine, as no distinction is made between biological sexes in dosing, even in routine clinical practice. The administration of a predefined amount of the local anesthetic also allowed obtaining comparable pharmacokinetic data. The final evaluation of ulnar nerve function and puncture site of the nerve block was performed seven days after the study via a telephone interview.

## Outcome variables

**Sensory blockade.** Sensory blockade was assessed by pinprick testing (100 = no difference, 0 = no sensory reception) at the hypothenar area in comparison with the contralateral side. Five areas of sensory supply were defined: dorsal side hypothenar muscles, ulnar side hypothenar area, palmar side hypothenar muscles, fifth finger, and ulnar side fourth finger. Short bevel needles were used for Pinprick testing, where the tip of the needle was applied with a force adequate to indent the skin without puncturing it, producing a consistent painful sensation when applied to non-blocked areas. Time points for sensory assessment were baseline, 2, 4, 6, 8, 10, 15, 20, 30, 60 min after the block, and then every 30 min until complete recovery of the nerve block.

*Onset of sensory blockade* was defined as pinprick = 0 in 4 from 5 of the described areas. *Duration of sensory blockade* was defined as pinprick testing = 80 in the respective hypothenar areas. We decided to define sensory block duration when pinprick testing = 80, rather than 100, to avoid extremely long durations of investigation.

**Pharmacokinetic analysis.** Samples of blood (4 mL) were drawn for the determination of plasma ropivacaine concentrations at baseline and 20, 40, 60, 120, 240 and 360 min. Blood samples were kept on ice for a maximum of 60 min before being centrifuged at 4°C and 2.600xg for 10 min to obtain plasma. Plasma samples were stored at -80°C until analyses. A mixture of 100.0 μL sample (plasma/calibrator/quality-control), 100.0 μL of phosphate buffer pH 7.4, and 10.0 μL internal standard solution (2500 ng.mL$^{-1}$ d$_7$-ropivacaine in methanol) was extracted with 200 μL of n-butyl chloride. The sample was vortexed gently for 5 min,

centrifuged at 20.800xg for 5 min, and 100 μL of the supernatant was transferred to a GC-MS vial. The GC-MS system consisted of a 7890B gas chromatograph coupled to a 7000C GC-MS/ MS Triple Quad mass spectrometer (Agilent, Santa Clara, CA, USA). An autosampler AS7693 served for splitless injection onto a HP-5ms Ultra Inert capillary column (30 m, 0.25 mm internal diameter, 0.25 μm film thickness; Agilent). Injector temperature was 280°C and injection volume was 2 μl. Helium was used as carrier gas at a flow rate of 1.6 mL.min$^{-1}$. Initial oven temperature was 140°C for 2 min, raised to 200°C with 25°C.min$^{-1}$, then to 300°C with 30°C. min$^{-1}$,and held for 3 min. Transfer line temperature was 300°C. After electron-impact (EI) ionization the mass spectrometer was operated in the multiple reaction monitoring (MRM) mode using the mass transitions m/z 126.0 to m/z 84.1 as quantifier and m/z 126.0 to m/z 98.1 and m/z 56.0, respectively as qualifiers for ropivacaine as well as m/z 133.0 to m/z 85.0 as quantifier for $d_7$-ropivacaine (internal standard). The calibration range for quantitative analyses was 5 ng.mL$^{-1}$ to 500 ng.mL$^{-1}$. MassHunter MS-Quantitative Analysis software was used for evaluation (Agilent, Santa Clara, CA, USA). The whole procedure was validated according to the European Medicines Agency (EMA) Bioanalytical Method Validation Guideline (EMEA/ CHMP/EWP/192217/2009) achieving a lower limit of quantification of 5 ng.mL$^{-1}$ for ropivacaine. The European Pharmacopoeia (EP) Reference Standard for ropivacaine from Sigma-Aldrich (St. Louis, Missouri, USA) was used in this study; $d_7$-ropivacaine was purchased from trc canada (North York, ON, Canada).

**Morphometric analysis of nerves.** The *visibility of ulnar nerves* in ultrasonography was investigated via grey-tone analysis of the ulnar nerve relative to the surrounding tissue. Prior to nerve blockade, a JPEG image of the nerve and the surrounding tissue was stored for subsequent grey-tone analysis with the Affinity Photo 2.0.4 for MAC image analysis software (Serif Europe Ltd, Nottingham, UK). The median value of grey-tone was evaluated of the ulnar nerve and the surrounding tissue in the 12, 3, 6 and 9 o´clock positions and the difference in grey-scales was calculated according to the formula: *Relative visibility of the nerve = Grey scale value nerve/(ΣGrey scale values (12 + 3 + 6 + 9 o'clock)/4)* [8]. In addition, the Vienna Scoring System [9, 10] for descriptive evaluation of peripheral nerves (1 = internal structure of the nerve visualized, 2 = the nerve is visualized as a circular or oval-shaped blight halo, the epineurium, 3 = the nerve is visualized as reflection determined by the anatomy of the surrounded tissue, 4 = the anatomical position of the nerve shows no response to the ultrasound beam, or isoechoic behaviour), was used to mark differences of echotexture.

## Power and statistical analysis

A previous study with ropivacaine 0.75%wt/vol for ulnar nerve blockade showed a duration of sensory blockade of 350 min with the largest SD of 54 minutes [11]. We expected a true difference of approximately 35 minutes (= 10%). Assuming a true difference in means between the test and the reference group of 35 minutes, a pooled standard deviation of 54 minutes, the study would require a sample size of 39 for each group (i.e. a total sample size of 76), to achieve a power of 80% and alpha error of 5% (two-sample Student *t*- statistic). For nonparametric analysis, the sample size was increased by 15% to 90 subjects.

After discussion with the Ethical Committee, an interim analysis based on the primary outcome parameter was to be performed after a pilot study of 12 subjects from each biological sex. This was to inform further sample size calculations in relation to initial calculated number of 90 cases (power = 80%, level of significance = 5%) and to find nominal differences of 10 to 15% in the primary outcome of duration of sensory blockade as significant.

Secondary outcomes included onset of sensory block, pharmacokinetic characteristics and visibility of the nerves. Plasma pharmacokinetic parameters were calculated with non-

compartmental analysis using a commercially available software program (Phoenix® Win-Nonlin®, Certara USA, Inc., Princeton, NJ, USA).

Data are presented as median [interquartiles] and count (%) and analysed using unpaired Mann-Whitney *U*- and Fisher's exact statistics. Significance was defined at *P*<0.05 (two-sided) with exact *P* values and differences are presented with 95% confidence intervals (CI). Additional analyses included analysis of covariance (ANCOVA) with robust standard errors to adjust for volunteer and baseline characteristics with adjusted *P* values. Linear mixed effects models with robust standard errors and identity covariance structures were also used to analyse ropivacaine plasma concentrations. Software used for statistical analysis included Number Cruncher Statistical Systems (NCSS) 2020, NCSS Inc., Kaysville, UT and Stata 17.0, StataCorp, College Station, TX and GraphPad Prism version 10.0.0, GraphPad Software, Boston, Massachusetts USA.

## Results

Based on the CONSORT statement Fig 1 illustrates the study roadmap. After the 24 volunteers were studied, we performed an updated power analysis based on the results of the primary outcome, duration of sensory blockade. Mean (SD) duration of blockade was 460.0 (53.3) min in females and 475.0 (52.5) min for male volunteers with a difference 15.0 min (95% CI -59.8, 29.8; *P* = 0.49). The required minimum sample sizes were estimated at 414 and 552 study participants after nonparametric adjustment, with powers of 80% and 90% respectively for this smaller difference at an alpha error of 5%. We therefore decided to terminate the study after these 24 subjects.

Volunteer characteristics are shown in Table 1. As expected, men were significantly taller and heavier than women and 83% of the blocks were performed on the left ulnar nerve.

Pharmacodynamic outcomes and morphometric analysis of nerves are presented in Table 2 and did not show significant differences between women and men. The median (IQR) duration of sensory blockade as the primary outcome was 450 (420; 503) min in women and 480 (450; 510) min in men (*P* = 0.49) with a median difference of 30 (95%CI -30, 60) min. The results were similar using ANCOVA after adjusting for age and weight of the subjects.

Pharmacokinetic parameters are presented in Table 3. Plasma ropivacaine concentrations were significantly higher in women by 33.5 ng mL$^{-1}$ (95%CI 0.3, 66.6; *P* = 0.048) over the 6 hour period. Statistically significant differences were found in half-life (*P* = 0.008), area under the concentration (AUC, calculated according to the trapezoidal rule) time curve to infinity (*P* = 0.017) and clearance (*P* = 0.018). ANCOVA adjustments for age and weight of the subjects confirmed shorter half-life, lower AUC to infinity and faster clearance in men.

We did not detect any side effects or complications seven days after nerve blockade. No adverse events occurred during the course of the study.

## Discussion

This is the first study investigating the impact of biological sex on peripheral nerve blockade. The study was terminated after an interim analysis of 24 study participants. Neither the duration of sensory blockade as primary outcome nor sensory onset time or ultrasound visibility of peripheral nerves as secondary outcomes showed significant differences. The total drug exposure across time, represented by the AUC to infinity, was significantly higher in women. Power calculations suggested that much larger studies would be required to find only marginal differences as possibly significant in the duration of sensory blockade as primary outcome parameter.

Biological sex is a relevant factor in medicine and needs to be considered in diagnosis and treatment. Specific findings in cardiology, representing the speciality at the forefront in sex-

## CONSORT 2010 Flow Diagram

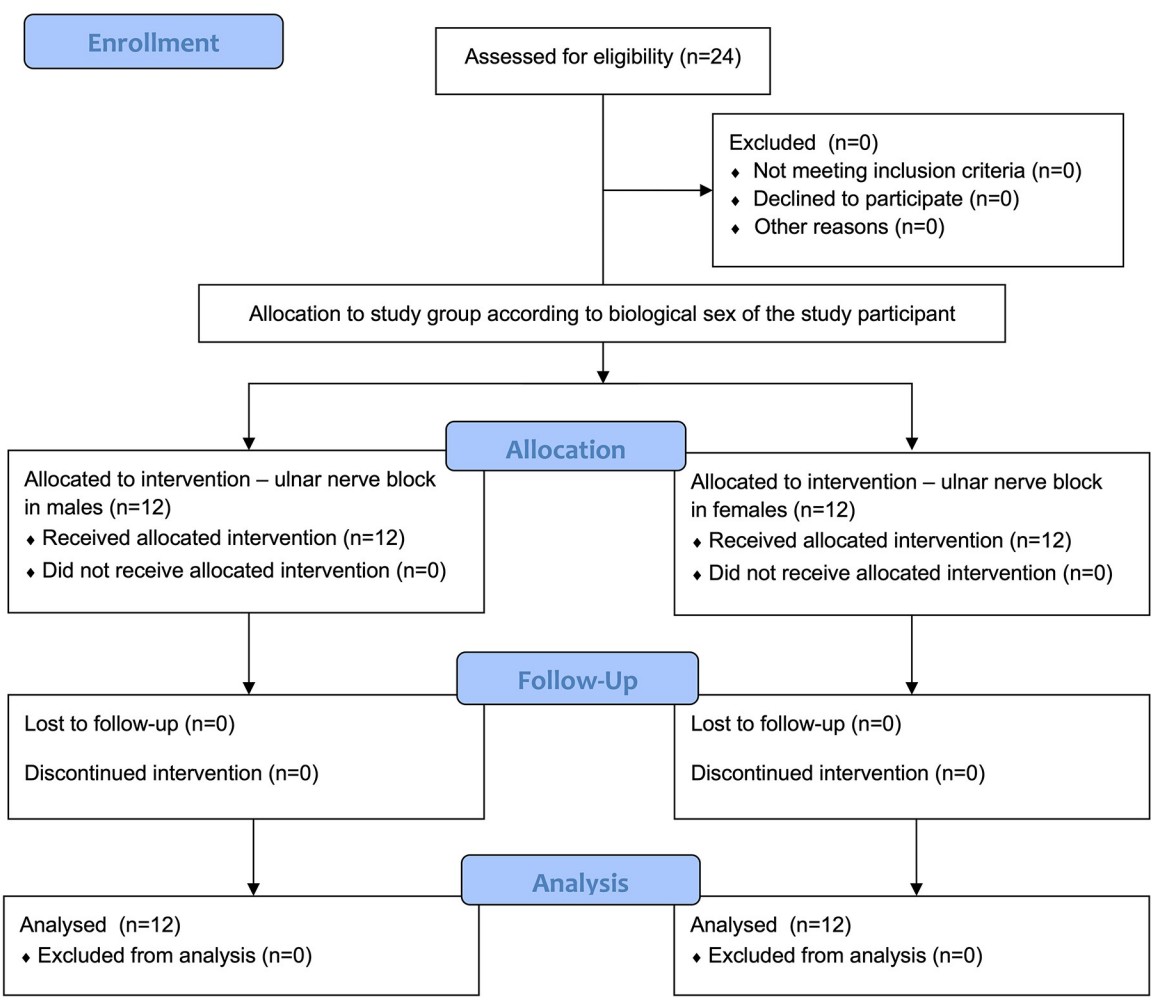

**Fig 1. Study flow diagram based on CONSORT.**

**Table 1. Volunteer characteristics.**

|  | Women | Men | *P*-value |
|---|---|---|---|
| **N** | 12 | 12 |  |
| **Age (years)** | 25.5 [23.3, 33.5] | 30.0 [28.0, 39.5] | 0.08 |
| **Weight (kg)** | 62.5 [56.0, 70.8] | 80.0 [66.3, 90.3] | 0.0064 |
| **Height (m)** | 1.67 [1.64, 1.73] | 1.78 [1.72, 1.84] | 0.0015 |
| BMI (kg m$^{-2}$) | 22.3 [20.3, 24.1] | 25.0 [22.4, 25.9] | 0.10 |
| **Side (left: right)** | 10:2 (left 83.3%) | 10:2 (left 83.3%) | 1.00 |

Data are presented as median [interquartiles] or count and were analysed using Mann-Whitney *U*- with exact *P*-values or Fisher's exact statistics as appropriate.

**Table 2. Primary and secondary outcomes.**

|  | Women | Men | Difference (95%CI) | *P*-Value | *P*-Adjusted |
|---|---|---|---|---|---|
| **Sensory duration (min)** | 450 [420, 503] | 480 [450, 510] | 30 (-30, 60) | 0.49 | 0.80 |
| **Sensory onset (min)** | 15 [10, 30] | 15 [8, 15] | -5 (-15, 2) | 0.19 | 0.21 |
| **Grey tone difference** | 42 [28, 55] | 47 [39, 63] | 9 (-7, 21) | 0.27 | 0.40 |
| **Vienna score** | 2 [1, 3] | 1 [1, 2] | 0 (-1, 0) | 0.45 | 0.76 |

Results are presented as median [interquartiles], Mann-Whitney *U*- statistics with exact *P*-values and median difference with 95% confidence interval (CI). Analysis of covariance was used to adjust for age and weight of subjects to estimate adjusted *P*-values.

specific science, indicate sex-differences in cardiovascular diseases and subsequent myocardial infarction [12], and in the diagnosis and treatment of men and women with type-2 diabetes where differences in endothelial dysfunction, atherosclerosis and coagulation were detected as outcome-relevant pathophysiological factors in the development of diabetic cardiomyopathy [13]. Another study in the field of cardiology shows that the interpretation of non-invasive measured blood pressure needs to be adapted to the biological sex, since the same non-invasive measured systolic blood pressure in women is associated with a higher invasive aortic systolic blood pressure as compared with men, which might be relevant for cardiovascular risk prediction [14]. Sex-related differences in blood coagulation were detected by Gorton et al. [15], and a recent study in the field of gastroenterology found sex-related differences in the development of gastroesophageal reflux disease [16].

The impact of biological sex in pharmacodynamics and pharmacokinetics in anesthesia is equally relevant, and a number of anesthetic drugs have demonstrated differences in this context [17]. Animal studies in pain perception and regional anesthesia showed controversial results regarding sex-related differences. Velichova et al. detected sex-dependent differences in C-fibre activity in rats [18], whereas Cullion et al. did not find sex-differences of nerve block duration in rats [6]. Human studies in this field are also controversial [5, 7, 19–21], but it is important to notice that no study was designed with the main focus of biological sex differences of pharmacodynamic and pharmacokinetic characteristics in peripheral regional anesthesia.

We therefore used our ulnar nerve block model in volunteers [22] to investigate the relationship of biological sex and pharmacodynamic and pharmacokinetic characteristics of peripheral nerve blockade. In addition, we investigated the biological sex dependent visibility

**Table 3. Pharmacokinetic parameters.**

|  | Women (n = 12) | Men (n = 12) | *P*-Value | *P*-Adjusted |
|---|---|---|---|---|
| $C_{max}$ **(ng/mL)** | 292.0 [248.7, 332.3] | 251.4.[201.8, 303.1] | 0.13 | 0.34 |
| $t_{max}$ **(h)** | 0.70 [0.30, 0.70] | 0.70 [0.70, 0.70] | 0.11 | 0.014 |
| $T_{1/2}$ **(h)** | 3.6 [2.9, 4.3] | 2.3 [2.0, 3.0] | 0.008 | 0.004 |
| $AUC_{0-6h}$ **(h\*ng/mL)** | 792.5 [731.8, 1136.1] | 698.7 [598.1, 901.8] | 0.052 | 0.14 |
| $AUC_{0-infinity}$ **(h\*ng/mL)** | 1164.2 [1014.4, 1934.4] | 841.8 [739.3, 1156.6] | 0.017 | 0.037 |
| $V_d$ **(L)** | 93.1 [79.2, 100.8] | 88.1 [80.5, 99.9] | 0.97 | 0.18 |
| **Cl (L/h)** | 19.4 [11.8, 22.2] | 26.7 [19.6, 30.6] | 0.018 | 0.034 |

Results are presented as median [interquartiles] and Mann-Whitney *U*- statistics with exact *P*-values. Analysis of covariance was used to adjust for age and weight of subjects to estimate adjusted *P*-values. $C_{max}$ = maximum concentration; $t_{max}$ = time to maximum concentration; $T_{1/2}$ = terminal elimination half-life; $AUC_{0-6h}$ = area under the concentration-time curve for the sampling period of 6 hours; $AUC_{0-infinity}$ = area under the concentration-time curve for the sampling period to infinity; $V_d$ = apparent distribution volume; Cl = clearance.

of the ulnar nerve in ultrasound, since previous studies in this field also have shown controversial results [23, 24]. A study in volunteers is particularly suitable to answer this question, because the reliability of sensory testing is better as compared with patients undergoing surgery, where additional factors (main focus on treatment, type of surgery, impairment of sensory testing due to postoperative casts) need to be considered. Volunteers are motivated and focused on an exact description of regional block characteristics without any surgery-related distraction. In addition, physicians performing the study are also exclusively focused on the exact realization of the study protocol.

The primary outcome, defined as duration of sensory block by pinprick testing, showed similar results between women and men (Table 2). As highlighted in the Methods section, we performed a planned power analysis after 24 volunteers. Based on the results of the primary outcome, 414 volunteers with a power of 80% and 552 volunteers with a power of 90% would be required for statistical significance. Since volunteer studies should be performed with a minimum of participants, to minimize study-associated side effects or risks, to reduce costs and to reduce human resources, a study with such large numbers of volunteers in the field of regional anesthesia is unrealistic to find marginal differences.

The ultrasonographic appearance of nerves is a crucial factor for the successful performance of regional nerve blocks. In 2014, we developed a method to quantify the ultrasound visibility of peripheral nerves via grey tone analysis [8]. Similar to pharmacokinetic outcomes in the present study, we did not detect significant biological sex dependent differences in the grey tone analysis of the ulnar nerve. This result is in accordance with a publication where a subjective scoring was performed in upper extremity nerve structures [24]. In addition to the mathematical model that we used, we performed an analysis using the "Vienna scoring system", which is based on a clear definition of nerve morphometry [9, 10]. This scoring system also did not show significant differences between women and men.

Contrary to pharmacodynamic and nerve-related morphometric characteristics, specific pharmacokinetic parameters of the local anesthetic (ropivacaine) showed sex-related differences (faster $t_{max}$, longer $T_{1/2}$, increased $AUC_{0\text{-infinity}}$ and slower Cl), thus indicating a faster increase of ropivacaine serum levels ($t_{max}$) and a greater total drug exposure across time ($AUC_{0\text{-infinity}}$). Men show a higher activity of Cytochrome P450 1A2 [25], which is the main monooxygenase catalyzing the metabolism of ropivacaine [26], thus explaining higher clearance and shorter half-life of ropivacaine. A possible consequence of the pharmacokinetic results of our study could be an increased sensitivity to local anesthetic intoxication of women compared with men. The clinical consequence of this hypothesis is not investigated in the literature, and subsequent studies need to focus on this topic.

The evaluation of the quality of regional blocks is a function of the regional block itself and the perception of pain. Perception of pain is a complex issue with differences between women and men [27, 28]. The present study investigated only the pharmacodynamic characteristics of nerve blocks, whereas the effects in patients, such as pain perception, needs an improved understanding and specific scientific efforts.

The present study is of particular importance for the general applicability of our past studies in volunteers, where we included only male study participants to exclude possible biological sex differences [11, 22, 29–31]. The similarity of pharmacodynamic and morphometric characteristics in the present study enables the extrapolation of the previous results to the female population.

In summary, the present study was terminated after an interim analysis of 24 study volunteers. The pharmacodynamic characteristics of nerve blocks and the ultrasound visibility of the ulnar nerve were similar in men and women. Pharmacokinetic characteristics showed sex-dependent differences. Based on the results of this study, biological sex is not a factor that

needs to be considered in peripheral regional anesthesia for pharmacodynamic outcomes, but it does influence pharmacokinetic characteristics with greater exposure for ropivacaine in women.

## Supporting information

**S1 Table. TREND statement checklist.** The checklist to standardize reporting of nonrandomized trials.
(PDF)

**S1 Protocol. Study protocol.**
(DOCX)

## Acknowledgments

We greatfully thank Maria Weber (certified study nurse) for her invaluable support throughout the study.

## Author Contributions

**Conceptualization:** Markus Zadrazil, Peter Marhofer, Malachy Columb, Philipp Opfermann, Werner Schmid, Daniela Marhofer, Thomas Stimpfl, Markus Zeitlinger.

**Data curation:** Markus Zadrazil, Peter Marhofer, Malachy Columb, Philipp Opfermann, Markus Zeitlinger.

**Formal analysis:** Markus Zadrazil, Peter Marhofer, Malachy Columb, Philipp Opfermann, Markus Zeitlinger.

**Investigation:** Markus Zadrazil, Peter Marhofer, Philipp Opfermann, Werner Schmid, Daniela Marhofer, Thomas Stimpfl, Valentin Al Jalali, Markus Zeitlinger.

**Methodology:** Peter Marhofer, Malachy Columb, Philipp Opfermann, Thomas Stimpfl, Sabine Reichel.

**Project administration:** Markus Zadrazil.

**Resources:** Peter Marhofer, Daniela Marhofer, Markus Zeitlinger.

**Software:** Malachy Columb.

**Supervision:** Markus Zadrazil, Peter Marhofer, Philipp Opfermann, Daniela Marhofer.

**Validation:** Markus Zadrazil, Peter Marhofer, Malachy Columb, Philipp Opfermann, Werner Schmid, Thomas Stimpfl, Sabine Reichel, Markus Zeitlinger.

**Writing – original draft:** Markus Zadrazil, Peter Marhofer, Malachy Columb, Philipp Opfermann, Markus Zeitlinger.

**Writing – review & editing:** Markus Zadrazil, Peter Marhofer, Philipp Opfermann, Markus Zeitlinger.

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
