## [Decision Letter · Decision Letter 0]

3 Dec 2023

PONE-D-23-35478The impact of biological sex in peripheral nerve blockade: A prospective pharmacodynamic, pharmacokinetic and morphometric study in volunteersPLOS ONE

Dear Dr. Zadrazil,

Thank you for submitting your manuscript to PLOS ONE. After careful consideration, we feel that it has merit but does not fully meet PLOS ONE’s publication criteria as it currently stands. Therefore, we invite you to submit a revised version of the manuscript that addresses the points raised during the review process.

The study involved volunteers in a clinical trial examining the influence of biological sex on peripheral nerve blockade's pharmacodynamic, pharmacokinetic, and morphometric characteristics. Minor revisions are suggested, including the replacement of "P<0.05" with significance levels in power calculations, specifying statistical methods for p-value estimation in the abstract, indicating the statistical testing method achieving 80% power, correcting a grammatical error related to Fisher's exact, stating the underlying covariance structure in linear mixed effects models, providing percentages for the left or right side in Table 1's "side" column, referencing the mathematical method for calculating AUC in line 219, adding sample sizes to the header row in Table 3, and reporting any adverse events during the study.

We look forward to receiving your revised manuscript.

Kind regards,

Lalit Gupta

Academic Editor

PLOS ONE

Journal Requirements:

2. We note that the original protocol file you uploaded contains a confidentiality notice indicating that the protocol may not be shared publicly or be published. Please note, however, that the PLOS Editorial Policy requires that the original protocol be published alongside your manuscript in the event of acceptance. Please note that should your paper be accepted, all content including the protocol will be published under the Creative Commons Attribution (CC BY) 4.0 license, which means that it will be freely available online, and any third party is permitted to access, download, copy, distribute, and use these materials in any way, even commercially, with proper attribution.

Therefore, we ask that you please seek permission from the study sponsor or body imposing the restriction on sharing this document to publish this protocol under CC BY 4.0 if your work is accepted. We kindly ask that you upload a formal statement signed by an institutional representative clarifying whether you will be able to comply with this policy. Additionally, please upload a clean copy of the protocol with the confidentiality notice (and any copyrighted institutional logos or signatures) removed.

Additional Editor Comments:

The study involved volunteers in a clinical trial examining the influence of biological sex on peripheral nerve blockade's pharmacodynamic, pharmacokinetic, and morphometric characteristics. Minor revisions were suggested, including the replacement of "P<0.05" with significance levels in power calculations, specifying statistical methods for p-value estimation in the abstract, indicating the statistical testing method achieving 80% power, correcting a grammatical error related to Fisher's exact, stating the underlying covariance structure in linear mixed effects models, providing percentages for the left or right side in Table 1's "side" column, referencing the mathematical method for calculating AUC in line 219, adding sample sizes to the header row in Table 3, and reporting any adverse events during the study.

Reviewers' comments:

Reviewer's Responses to Questions

**Comments to the Author**

1. Is the manuscript technically sound, and do the data support the conclusions?

Reviewer #1: Yes

Reviewer #2: Yes

2. Has the statistical analysis been performed appropriately and rigorously? 

Reviewer #1: Yes

Reviewer #2: Yes

3. Have the authors made all data underlying the findings in their manuscript fully available?

Reviewer #1: Yes

Reviewer #2: Yes

4. Is the manuscript presented in an intelligible fashion and written in standard English?

Reviewer #1: Yes

Reviewer #2: Yes

5. Review Comments to the Author

Reviewer #1: Volunteers participated in a clinical trial which aimed to investigate the impact of biological sex on pharmacodynamic, pharmacokinetic and morphometric characteristics for peripheral nerve blockade.

Minor revisions:

1- Abstract and line 197: Power calculations include significance levels rather than p-values. Replace “P<0.05” at all occurrences.

2- Abstract: Briefly state the statistical methods from which p-values were estimated.

3- Line 163: State the statistical testing method which achieves 80% power. Perhaps it is the t-test.

4- Lines 182 and 204: Grammatical error: Fisher’s exact.

5- Line 185: State the underlying covariance structure used in the linear mixed effects models and the criteria for selecting it.

6- Table 1: For side, provide the percentage corresponding to the left or right side.

7- Line 219: Provide a reference for the mathematical method used to calculate AUC.

8- Table 3: Include sample sizes in the header row.

9- Indicate if any adverse events occurred during the course of the study.

Reviewer #2: In the age of personalized medicine all details are important that makes a difference between individuals or interindividual differences can be excluded. This is the first human study investigating the impact of biological sex in peripheral nerve blockade. Regional anesthesia plays an important role not only during the surgery but in the postoperative period as well, and in other areas of medicine, including malignancies and obstetrics. Most of amid local anesthetics (including ropivacaine) are highly cardiotoxic compounds. The knowledge that higher peak serum levels may be expected in females may have importance in long term therapy using epidural catheters.

6. PLOS authors have the option to publish the peer review history of their article (what does this mean?). If published, this will include your full peer review and any attached files.

Reviewer #1: No

Reviewer #2: **Yes: **István Bátai

---

## [Author Response · Author response to Decision Letter 0]

11 Dec 2023

Please find our point-to-point answers to each point raised by the reviewers 

concerning the manuscript for the research article with the title “The impact of biological sex in 

peripheral nerve blockade: A prospective pharmacodynamic, pharmacokinetic and morphometric 

study in volunteers”. 

Reviewer #1: Volunteers participated in a clinical trial which aimed to 

investigate the impact of biological sex on pharmacodynamic, 

pharmacokinetic and morphometric characteristics for peripheral nerve 

blockade. 

Minor revisions: 

1- Abstract and line 197: Power calculations include significance levels 

rather than p-values. Replace “P<0.05” at all occurrences. 

• Thank you very much for this comment. It is correct in that ‘’P<0.05’’ should be replaced in 

power calculations and that this is specified better by the term ‘’alpha error of 5%’’, which 

has been changed throughout as suggested. 

2- Abstract: Briefly state the statistical methods from which p-values were estimated. 

• Mann-Whitney U-statistics were used for the data in the abstract and this has been added as 

suggested (line 34 and 35). 

3- Line 163: State the statistical testing method which achieves 80% power. Perhaps it is the t-test. 

• As requested by the reviewer, the unpaired Student t-statistic has been inserted for the 

power calculation. 

4- Lines 182 and 204: Grammatical error: Fisher’s exact. 

• The grammatical error has been corrected. 

5- Line 185: State the underlying covariance structure used in the linear 

mixed effects models and the criteria for selecting it. 

• In the used statistical software Stata, the covariance matrix is defaults to ‘’Identity’’ from 

others for single-variable random-effects specifications in subjects. This has now been added 

as suggested for completeness by the reviewer. 

6- Table 1: For side, provide the percentage corresponding to the left or right side. 

• For side, the percentage for the left side has been added in table 1 (left 83.3%). 

7- Line 219: Provide a reference for the mathematical method used to calculate AUC. 

• The mathematical method used was the trapezoid rule. The following text was inserted in 

the manuscript “AUC, calculated according to the trapezoidal rule”. 

8- Table 3: Include sample sizes in the header row. 

• The sample sizes (Women, n=12 and Men, n=12) have been added in the header of table 3. 

9- Indicate if any adverse events occurred during the course of the study. 

• It has been indicated, that no adverse events occurred during the course of the study – line 

232/233.

Reviewer #2: In the age of personalized medicine all details are important 

that makes a difference between individuals or interindividual differences 

can be excluded. This is the first human study investigating the impact of 

biological sex in peripheral nerve blockade. Regional anesthesia plays an 

important role not only during the surgery but in the postoperative period 

as well, and in other areas of medicine, including malignancies and 

obstetrics. Most of amid local anesthetics (including ropivacaine) are 

highly cardiotoxic compounds. The knowledge that higher peak serum levels 

may be expected in females may have importance in long term therapy using 

epidural catheters. 

• Thank you very much for this remark, appreciating our work.

---

## [Decision Letter · Decision Letter 1]

27 Dec 2023

The impact of biological sex in peripheral nerve blockade: A prospective pharmacodynamic, pharmacokinetic and morphometric study in volunteers

PONE-D-23-35478R1

Dear Dr. Zadrazil,

We’re pleased to inform you that your manuscript has been judged scientifically suitable for publication and will be formally accepted for publication once it meets all outstanding technical requirements.

Kind regards,

Lalit Gupta

Academic Editor

PLOS ONE

Additional Editor Comments (optional):

All the comments raised by the previous reviewer have been thoroughly addressed.

Upon careful reevaluation of the manuscript, it is evident that the revised version has significantly improved and is now deemed acceptable for acceptance.

Reviewers' comments:

Reviewer's Responses to Questions

**Comments to the Author**

1. If the authors have adequately addressed your comments raised in a previous round of review and you feel that this manuscript is now acceptable for publication, you may indicate that here to bypass the “Comments to the Author” section, enter your conflict of interest statement in the “Confidential to Editor” section, and submit your "Accept" recommendation.

Reviewer #1: All comments have been addressed

2. Is the manuscript technically sound, and do the data support the conclusions?

Reviewer #1: (No Response)

3. Has the statistical analysis been performed appropriately and rigorously? 

Reviewer #1: (No Response)

4. Have the authors made all data underlying the findings in their manuscript fully available?

Reviewer #1: (No Response)

5. Is the manuscript presented in an intelligible fashion and written in standard English?

Reviewer #1: (No Response)

6. Review Comments to the Author

Reviewer #1: (No Response)

7. PLOS authors have the option to publish the peer review history of their article (what does this mean?). If published, this will include your full peer review and any attached files.

Reviewer #1: No

---

## [Editor Report · Acceptance letter]

15 Jan 2024

PONE-D-23-35478R1 

PLOS ONE

Dear Dr. Zadrazil, 

I'm pleased to inform you that your manuscript has been deemed suitable for publication in PLOS ONE. Congratulations! Your manuscript is now being handed over to our production team.

Kind regards, 

on behalf of

Dr. Lalit Gupta 

Academic Editor

PLOS ONE